# Forecasting Postoperative Delirium in Older Adult Patients with Fast-and-Frugal Decision Trees

**DOI:** 10.3390/jcm11195629

**Published:** 2022-09-24

**Authors:** Maria Heinrich, Jan K. Woike, Claudia D. Spies, Odette Wegwarth

**Affiliations:** 1Department of Anesthesiology and Operative Intensive Care Medicine (CCM, CVK), Charité-Universitätsmedizin Berlin, Corporate Member of Freie Universität Berlin, Humboldt-Universität zu Berlin, and Berlin Institute of Health, 13353 Berlin, Germany; 2Berlin Institute of Health@Charité (BIH), Anna-Louisa-Karsch 2, 10178 Berlin, Germany; 3School of Psychology, University of Plymouth, Plymouth PL4 8AA, UK; 4Max Planck Institute for Human Development, Center for Adaptive Rationality, 14195 Berlin, Germany; 5Heisenberg Chair for Medical Risk Literacy and Evidence-Based Decisions, Charité-Universitätsmedizin Berlin, Corporate Member of Freie Universität Berlin, Humboldt-Universität zu Berlin, and Berlin Institute of Health, 10117 Berlin, Germany

**Keywords:** fast-and-frugal decision trees, postoperative outcomes, postoperative delirium, clinical data prediction, medical decision making

## Abstract

Postoperative delirium (POD) is associated with increased complication and mortality rates, particularly among older adult patients. However, guideline recommendations for POD detection and management are poorly implemented. Fast-and-frugal trees (FFTrees), which are simple prediction algorithms, may be useful in this context. We compared the capacity of simple FFTrees with two more complex models—namely, unconstrained classification trees (UDTs) and logistic regression (LogReg)—for the prediction of POD among older surgical patients in the perioperative setting. Models were trained and tested on the European BioCog project clinical dataset. Based on the entire dataset, two different FFTrees were developed for the pre-operative and postoperative settings. Within the pre-operative setting, FFTrees outperformed the more complex UDT algorithm with respect to predictive balanced accuracy, nearing the prediction level of the logistic regression. Within the postoperative setting, FFTrees outperformed both complex models. Applying the best-performing algorithms to the full datasets, we proposed an FFTree using four cues (Charlson Comorbidity Index (CCI), site of surgery, physical status and frailty status) for the pre-operative setting and an FFTree containing only three cues (duration of anesthesia, age and CCI) for the postoperative setting. Given that both FFTrees contained considerably fewer criteria, which can be easily memorized and applied by health professionals in daily routine, FFTrees could help identify patients requiring intensified POD screening.

## 1. Introduction

Postoperative delirium (POD) is an acute and sudden change in the mental state, characterized by fluctuating levels of attention, consciousness and cognition [1,2]. The occurrence of POD is associated with increased complication and mortality rates [3,4] and may be related to the development of long-term cognitive disorders [5,6,7]. Incidence depends on predisposing and precipitating risk factors [8,9,10] and ranges from 10–50% [11,12], and older people are particularly susceptible to POD [13].

Given the risks associated with undetected postoperative delirium, it is important to have tools available to detect POD reliably and in a timely manner. According to the recommendations of the evidence-based and consensus-based guidelines on postoperative delirium [11], screening for delirium should be performed once per shift, at least twice a day, for 5 days after surgery in all patients, and predisposing and precipitating risk factors should be attenuated whenever possible. However, the implementation of these measures requires a considerable allocation of personnel resources and time, and it may thus come as no surprise that these guideline recommendations are poorly implemented [14]. Furthermore, a number of predictive models have been developed in the past to guide the prediction of POD; however, these also often require extensive assessment, and their clinical implications remain unclear [15,16,17,18,19,20]. These examples suggest that a detection tool that would be supportive of use in clinical care needs to be simple to keep the number of personnel and the time costs of the assessment as low as possible. Fast-and-frugal trees (FFTrees)—binarizing prediction algorithms based on limited information search—can provide such a simple structure and have demonstrated the capacity to facilitate accurate decisions in a variety of medical domains [21,22,23,24,25]. For instance, when predicting whether a patient presenting with chest pain should be admitted to the coronary care unit or to a normal ward, an FFTree consisting of only three yes-or-no questions performed comparably with a dedicated decision support tool (heart disease predictive instrument (HDPI)) requiring 50 pieces of information. These findings likely go against the common assumption that “more information is always better”, particularly in the medical domain, where most professionals may feel that, to make a good prediction or diagnosis, gathering more rather than less information reduces the risk of error. However, the relation between the amount of information and the quality of prediction is often an inverse U-shaped curve [26,27], specifically when situational uncertainty is high, as is the case in most medical situations including the prediction of POD. When situational uncertainty is high, model robustness is key [28,29]. Complex models, by using as much information as possible, fit “noise” and idiosyncrasies in the presented dataset that do not generalize to a new sample of patients. The result is “overfitting”, which conflicts with the robustness of a model and, thus, with the accuracy of prediction. Furthermore, it is important to note that the POD risk detected at admission (predisposing factors) increases substantially during the operation, and the impact of anesthesia and surgery (precipitating factors, such as trauma, stress, medication, depth of anesthesia, blood pressure fluctuations, transfusions) warrant the reassessment of risk. This means that models for POD must be adaptable and must include the conditions associated with surgery.

The aim of this work was to examine if FFTrees are able to sufficiently predict POD. To address the requirements of perioperative medicine, we built a pre-operative FFTree based on pre-operative parameters and further built a postoperative FFTree with modeling that additionally considered intraoperative parameters. Moreover, we compared the ability to predict unseen cases in the two FFTree construction methods [22]—which are based on limited information search—with those of two compensatory models; namely, unconstrained classification trees (based on the classification and regression trees (CART) algorithm) and logistic regression.

## 2. Materials and Methods

### 2.1. Overview of the Present Study

The work reported herein was performed on data initially acquired via the BioCog project, a prospective multicenter observational study conducted at the Charité–Universitätsmedizin Berlin, Department of Anesthesiology and Operative Intensive Care Medicine, Berlin, Germany, and the University Medical Center Utrecht, Department of Intensive Care Medicine, Utrecht, the Netherlands. This work was a secondary analysis performed for the purpose of generating FFTrees considering various influencing variables from the BioCog database generated at the study site Charité–Virchow Klinikum (n = 394, see Figure 1) in relation to the development of POD. The secondary analysis was approved by the local Ethics Committee (ref: EA2_048_18, 16 July 2020) and conducted in accordance with the Declaration of Helsinki.

The BioCog dataset was based on patients aged ≥ 65 years who were scheduled for elective surgery and presented with a Mini-Mental-State-Examination (MMSE) score of 23 points or higher (for detailed inclusion and exclusion criteria, see [30]).

### 2.2. Assessment of Postoperative Delirium

The models of this work inferred whether each respondent was at risk of POD as defined by the criteria of the fifth edition of the Diagnostic and Statistical Manual of Mental Disorders (DSM-5) [31]. Patients were considered delirious if they met any one of the following criteria:≥2 cumulative points on the Nursing Delirium Screening Scale (Nu-DESC) and/or a positive Confusion Assessment Method (CAM) score;a positive CAM score for the Intensive Care Unit (CAM-ICU);a patient chart review that showed descriptions of delirium (e.g., confused, agitated, drowsy, disorientated, delirious, received antipsychotic therapy).

Delirium screening was started in the recovery room and repeated twice per day at 08:00 and 19:00 (±1 h) for up to seven days after surgery. Delirium assessment was conducted independently of the routine hospital procedures by a research team that was trained and supervised by psychiatrists and delirium experts.

### 2.3. Cues

We aimed to develop two different models predicting a patient’s POD status based on (i) pre-operative cues alone and (ii) both pre-operative and intraoperative cues.

For the pre-operative model, each model was meant to categorize a patient as being or not being at risk of POD based on the following cues: age; sex; body height; body mass index; physical status according to the American Society of Anesthesiologists (ASA PS); Charlson Comorbidity Index (CCI) [32]; comorbidities, such as arterial hypertension, coronary artery disease, diabetes mellitus, stroke or transient ischemic attack, in medical history; education according to International Standard Classification of Education (ISCED) [33]; MMSE; pre-operative cognitive impairment (for details, see Appendix A); impaired activities of daily living according to Barthel (ADL) [34], as well as Lawton and Brody (IADL) [35]; malnutrition according to the Mini-Nutritional Assessment (MNA) [36]; pre-operative frailty status (for details, see Appendix A); depression according to the geriatric depression scale (GDS) [37,38]; pre-operative long-term medication with benzodiazepines; hazardous alcohol consumption based on the AUDIT score [39]; current smoker status; pack years and site of surgery (intracranial vs. intrathoracic, intra-abdominal or pelvic vs. peripheral). The postoperative model used, in addition to these pre-operative pieces of information, the duration of anaesthesia and the administration of premedication before surgery (benzodiazepines, clonidine, antihistaminergics, etc.).

### 2.4. Model Comparison

Two FFTree construction algorithms (the ifan algorithm (FFTi) and the dfan algorithm (FFTd)) [22] were compared with logistic regression and an unconstrained classification tree algorithm (UDT) based on CART [39] for the pre-operative and the complete dataset separately. We chose a maximum number of five cues for the FFTi algorithm and a maximum number of four cues for the FFTd algorithm. The criteria for ifan and dfan were set to balanced accuracy. For UDT, we weighted misclassifications of positive cases higher than the misclassification of negative cases (based on the ratio of negative to positive cases in the training set) to aim for good performance in terms of balanced accuracy. We used the rpart package in R for UDT [40], which implements most of the CART algorithms [39] (with the minimum splitting size set to 20 and the complexity parameter to 0.00001). For the binary logistic regression model (LogReg), cues that were provided to the corresponding tree models were included in the regression model. To target the criterion of balanced accuracy, we set the threshold to transform probability estimates into predictions to the base rate observed in the training set. We used the implementation of logistic regression in the glm command in R.

#### 2.4.1. Training and Test Set

In order to estimate the predictive performance of each model, the dataset was repeatedly randomly split into training and test (prediction set) sets, with an equal number of cases in each. Trees were constructed and parameters estimated based on the training set, and performance was measured based on the test set alone. The performance measure used was balanced accuracy—the mean of sensitivity and specificity—and models were estimated with the aim of achieving high values on this measure. Based on the model comparison, a tree construction algorithm was chosen to build two final trees (pre- and postoperative) based on the full dataset (n = 394). All analyses were run in R (version 4.1.2) [41].

#### 2.4.2. Model Comparison Procedure

In preparing the dataset, missing values were replaced before starting the model comparison. For the following variables, a missing value was replaced by the sample median: ISCED, GDS, pack years, duration of anaesthesia. For categorical variables, missing values were replaced by the mode, which was 0 (no impairment) in the case of arterial hypertension, coronary artery disease, diabetes mellitus, stroke or transient ischemic attack in medical history, pre-operative cognitive impairment, ADL, IADL and the administration of premedication before surgery and 3 (no impairment) in the case of MNA.

In each trial of the model comparison, the full dataset was randomly split into training and test sets with an equal number of cases (n = 197). Models were estimated using the training set and performance (sensitivity, specificity and balanced accuracy) was measured using the test set alone. This procedure was repeated 1000 times for the more time-intensive FFTree construction algorithm (FFTd) and 10,000 times for all others. In each trial, the same training-test split was applied for each model, with FFTd being restricted to the first 1000 splits.

## 3. Results

### 3.1. Patient Characteristics

Altogether, we used data for 394 older adult surgical patients; 99 patients (25.1%) fulfilled the criteria for POD (see Table 1 for patient characteristics).

### 3.2. Model Comparison

For each model comparison, we report the mean performance of the four algorithms for the training and test sets and present visualizations of the distribution of prediction results across trials.

#### 3.2.1. Performance of Pre-Operative Models

The model performance for all four models is summarized in Table 2. Performance for the training set represents the ability to predict criterion values based on already known cue values. The table reports sensitivity, specificity and balanced accuracy (the average of sensitivity and specificity).

Unconstrained decision trees exhibited the best average balanced accuracy (0.803), followed by logistic regression and the two FFTree models. The difference between the best and worst model was over 0.11. The standard error for FFTd models was higher due to the smaller number of trials (1000 vs. 10,000). It should be noted that the standard error for balanced accuracy was lower than that for sensitivity and specificity: Models tended to trade off sensitivity against specificity across trials, resulting in more stable values for the average. The performance of unconstrained decision trees suffered the most, changing the order of performance in the test set.

The distribution of results across trials (see Figure 2) demonstrated the variability across trials and put the average differences into context. Further analysis showed that the LogReg model outperformed the FFTi model in 76.0% of the trials, but it was outperformed by the FFTi model in 23.9% of the trials.

#### 3.2.2. Performance of Postoperative Models

Adding the postoperative variables improved the performance of all models, both in fitting the training set and predicting the test set (see Table 3).

Again, the order of performance changed between fitting and prediction, with unconstrained decision trees showing the best fitting performance (0.840) and the worst prediction performance (0.660). In contrast to the previous comparison, the FFTrees outperformed both alternative models in prediction. The FFTd algorithm showed a better average balanced accuracy in prediction (0.704) than the FFTi algorithm (0.696). FFTi performed better than LogReg in 56.1% of the trials, and FFTd performed better than LogReg in 61.9% of the trials and better than FFTi in 56.2% of the trials. The distribution of balanced accuracy in the prediction task is shown in Figure 3. Thus, these differences were not generated by outliers but by a general tendency to outperform the competitors.

### 3.3. Decision Trees Based on Full Dataset

Based on the results of the model comparison, we estimated two fast-and-frugal trees based on the full dataset, one for the pre-operative set of variables and one for the complete, postoperative set of variables. We present visualizations of the resulting trees and report performance statistics based on the full dataset (n = 394).

#### 3.3.1. Pre-Operative FFTree

The first (pre-operative decision tree) was based on pre-operative data. Following the results of the model comparison, we chose the ifan algorithm to construct the tree. The resulting pre-operative decision tree contained four cues (CCI, site of surgery, ASA PS and frailty status) and indicated a sensitivity of 0.84 and a specificity of 0.46 with a balanced accuracy of 0.65 (see Figure 4). On average, 1.8 cue values had to be looked up to make a decision for the cases in the dataset, and 93% of the provided information was ignored on average. Over 52% of all cases were classified as positive after the first question (with a CCI larger than 1), and 71 of the 99 positive cases were in this group. The tree achieved a higher sensitivity at the cost of specificity (the unweighted accuracy was 0.56).

#### 3.3.2. Postoperative FFTree

The construction of the second decision tree (postoperative decision tree) took intraoperative parameters into account in addition to the pre-operative data. Following the results of the model comparison, we chose the dfan algorithm for tree construction in this case. The postoperative decision tree contained three cues (duration of anesthesia, age and CCI). While the maximum depth was set to four cues, the algorithm did not find an improvement by adding an additional layer to the tree, generating a truncated tree (see also [29]).

The decision tree demonstrated a sensitivity of 0.81, a specificity of 0.72 and a balanced accuracy of 0.76 (see Figure 5). It used 2.1 cues on average to make a classification, and it ignored 92% of the provided information on average. The tree was more balanced than the pre-operative tree and achieved an unweighted accuracy of 0.74.

## 4. Discussion

The aim of this work was to develop decision trees that can be used to estimate the risk of developing POD both pre-operatively and postoperatively in older adult patients. We were able to create two decision trees that differed in the parameters included. The pre-operative decision tree contained four cues (CCI, site of surgery, ASA PS and frailty status) and the postoperative contained three cues (duration of anesthesia, age and CCI). Before estimating fast-and-frugal trees (FFTrees), we compared two methods of FFTree construction (the ifan algorithm (FFTi) and the dfan algorithm (FFTd)) with unconstrained classification trees (UDTs, based on CART) and logistic regression. Fast-and-frugal trees are minimal binary classification trees that are constrained in terms of their structure. Various algorithms have been proposed for the construction of fast-and-frugal trees [22,25,29,42]. Here, we chose two algorithms that have been proved most competitive in achieving a high balanced accuracy [22], the ifan algorithm (FFTi) and the dfan algorithm (FFTd). A natural comparison for highly constrained fast-and-frugal trees are unconstrained classification trees (UDTs). Furthermore, we compared FFTrees with binary logistic regression models (LogReg), which predict the probability of an older adult patient being at risk for POD based on a weighted integration of all provided cues. It should be noted that, due to the relatively smaller number of positive cases in the sample, models aimed at achieving a high unweighted accuracy would, in contrast, likely sacrifice sensitivity for specificity, which would not be in line with the aims for the decision tool.

The results were in line with previous model comparisons [22,25,29]. More flexible models generally outperform less flexible models in this type of fitting performance. In line with this, unconstrained decision trees exhibited the best average balanced accuracy in the training set in the pre-operative model comparison, followed by logistic regression and the two FFTree models. The distribution of results across trials demonstrated the variability across trials and put the average differences into context. In the testing set with pre-operative modeling, logistic regression showed the highest balanced accuracy, closely followed by fast-and-frugal trees constructed with the ifan algorithm. All models showed worse performance in the testing set when predicting cases that were not part of the sample used to estimate their parameters. The inflation of predictive accuracy when predicting familiar cases has also been termed “overfitting”, and it is usually more pronounced in more complex and flexible models. As expected, the performance of unconstrained decision trees suffered the most, changing the order of performance in the testing set.

Adding the postoperative variables improved the performance of all models, both in fitting the training set and predicting the test set. Again, the order of performance changed between fitting and prediction, with unconstrained decision trees showing the best fitting performance and the worst prediction performance. In contrast to the previous comparison, the fast-and-frugal trees outperformed both alternative models in prediction. Based on these model comparisons, we chose the ifan algorithm for pre-operative testing and the dfan algorithm for FFTree construction. It should be noted that the model comparison used 50% of the full dataset, providing a training sample that was smaller than the full sample. The advantage of this method is that training samples were less correlated. However, larger sample sizes tend to make logistic regression and CART more competitive. Based on our results for model comparison, we argue that FFTrees performed similarly to logistic regression and were not necessarily superior. From our point of view, the choice of method could be guided by the availability of data: the more data there are, the better the case for using more complex methods to achieve better predictive capability. In smaller (and often typical) datasets, the case is stronger for FFTrees. Moreover, FFTrees are simpler to apply, easier to communicate and requires less information, but they are competitive in predicting cases.

For all the cues included in the FFTrees, there is strong evidence in the literature [11,43,44,45] that they are independent risk factors in the development of POD and have to be considered in perioperative care according to guideline recommendations. The fact that age was not considered in the pre-operative decision tree may reflect the relevance of biological age rather than chronological age. In the pre-operative decision trees, this is represented by frailty status. Duration of anesthesia had a strong impact in decision tree development. It can be regarded as a surrogate for extent of surgery and associated inflammation, toxicity of anesthesia or intraoperative complications, such as bleeding or organ damage. All of these factors influence the risk for developing POD. The evaluation of this simple surrogate (duration of anesthesia) is significant for clinical applicability.

To the best of our knowledge, this is the first time decision trees were created for the risk stratification of POD. In the past, more emphasis has been placed on developing predictive models with clinical implications that remain unclear. A number of prediction models have been developed but primarily to predict delirium risk (not POD risk) [17] or for ICU patients [15,18,20,46]. A simple translation to surgical patients is problematic, as the needs of surgical patients are not addressed. Although surgical patients have a baseline risk of developing POD, surgery is such a relevant incident that it requires reassessment of risk. This means that models for POD must be adaptable and must include the conditions associated with surgery.

The oldest prediction model for delirium, developed in the 1990s, included the evaluation of vision and cognitive impairment, severe illness and high urea nitrogen/creatinine ratio [17]. Prediction models for ICU delirium include—besides age parameters, which are primarily related to intensive care treatment, such as for coma—use of sedatives and morphine [46], respiratory failure [20], vasoactive medication use and requirement of continuous renal replacement therapy and mechanical ventilation [15]. Some of them were developed through retrospective analysis [15,18].

There are three noteworthy studies that address modeling for POD risk prediction that have been recently published. A nine-item model for predicting POD risk with an area under the curve (AUC) of 0.77 was developed in a cohort of patients with acute hip fracture [47]. In this model, ASA PS was also considered, as well as functional dependence and pre-operative use of mobility aids, which are surrogates for frailty. Although this model was based on an extensive dataset, there was a crucial limitation, as POD was not determined prospectively but retrospectively by means of a chart review. Another seven-item model with an AUC of 0.82 was developed in older adult orthopedic patients in the ICU [19]. POD was determined prospectively, which reduced the sample size accordingly. Here, intraoperative parameters could be considered. For example, in addition to age, major hemorrhage was also included in the score. This score appears promising but also has the limitations that it is a very specific patient group, not all of the parameters are routine parameters (level of interleukin-6) and pre-operative application of the score is not appropriate (three intraoperative parameters are included). These scores have in common the limitation that eight or nine criteria may be too extensive for routine daily use. In this regard, a score based on four items with an AUC of 0.83 for cardiac surgery patients is more feasible for clinical application [16]. This score includes age, evaluation of MMSE, insomnia needing medical treatment and low physical activity, which is equivalent to one item in our frailty definition. Here, an automatic calculator, which calculates the risk, is actually available. Nevertheless, intraoperative parameters are not considered in this approach either.

We were able to solve this challenge by developing both a tree for pre-operative use and a tree for postoperative use. Furthermore, the datasets for development of the decision trees included both parameters for which there is strong evidence regarding the association with POD and parameters for which there are only hypotheses. With our work, we were able to select the most relevant parameters and rank them. It has been shown that only parameters with existing strong evidence were relevant in our analysis. Sensitivity appeared to be adequate in both trees. Specificity was very low in the pre-operative tree at 0.46. This does not matter considering how these trees will be applied in clinical care. In theory, all patients should receive a complete delirium screening and predisposing and precipitating factors should be anticipated. Since this is not implemented across the board for various reasons, the application of decision trees was intended to fill this gap, at least for those patients who should receive screening and special attention in any case. There is no disadvantage in giving special attention pre-operatively to a probably higher number of patients with misclassified increased risk of delirium. The postoperative decision tree has a higher specificity, which is relevant for clinical application, since POD screening requires personnel resources that are supposed to be extensive. The next steps include validation of the decision trees and verification of clinical practicability.

### Strength and Limitations

A key strength of this study is the prospective design of the POD assessment. POD was assessed through a comprehensive, standardized and validated assessment. While the use of routine data for modeling, as in many of the previously discussed models, has the great advantage that extensive datasets are available, this must be viewed very critically, especially in the case of the clinical picture of postoperative delirium. Especially due to its fluctuating characteristics and the frequent occurrence of hypoactive forms, a comprehensive validated screening is essential. As described above, there is a large gap in POD coverage, so it cannot be assumed that POD screening is implemented adequately. This raises doubts about the quality of the analysis of routine data.

The study database contained extensive information on both parameters for which there is strong evidence regarding the association with POD and parameters for which there are only hypotheses. In addition, we were able to develop the decision trees based on a dataset of patients covering a wide range of surgical disciplines (see Appendix A), which reflects the conditions that apply to the perioperative risk evaluation settings in clinical practice and gives the translation approach a bit more feasibility.

Nevertheless, some important limitations must be considered. Even though our dataset was very high quality and extensive, the sample size was small. Therefore, the model comparison used 50% of the full dataset, providing a training sample that was smaller than the full sample. Larger sample sizes tend to make unconstrained classification trees and logistic regression more competitive whereas smaller sample sizes do the opposite. Therefore we would like to argue that the choice of method should be guided by the availability of data: the more data there are, the better the case for using more complex methods to achieve a better predictive capability. In smaller (and often typical) datasets, the case is stronger for the FFTrees. Finally, we were able to provide initial insights with our analysis, but these still need to be validated.

## 5. Conclusions

Within the pre-operative setting, FFTrees outperformed the more complex UDT algorithm with respect to their predictive balanced accuracy, nearing the prediction level of logistic regression. Within the postoperative setting, FFTrees outperformed both complex models. Applying the best-performing algorithms to the full datasets we propose an FFTree using four cues (CCI, site of surgery, ASA PS and frailty status) for the pre-operative setting and an FFTree containing only three cues (duration of anesthesia, age and CCI) for the postoperative setting. Given that both FFTrees contain considerably fewer criteria, which can be easily memorized and applied by health professionals in daily routine, FFTrees could help identify patients requiring intensified POD screening.

## Figures and Tables

**Figure 1 jcm-11-05629-f001:**
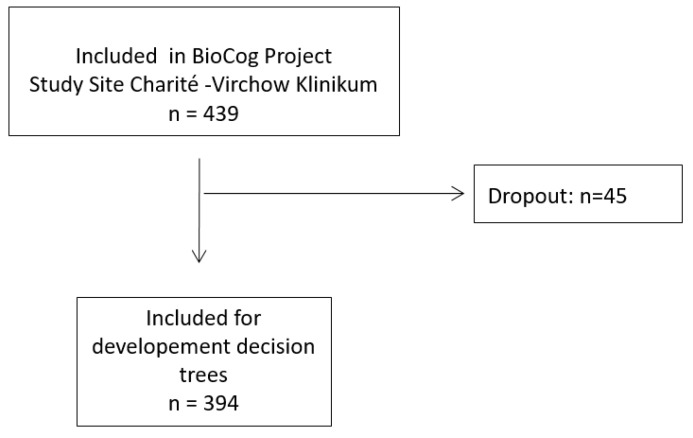
Flow chart.

**Figure 2 jcm-11-05629-f002:**
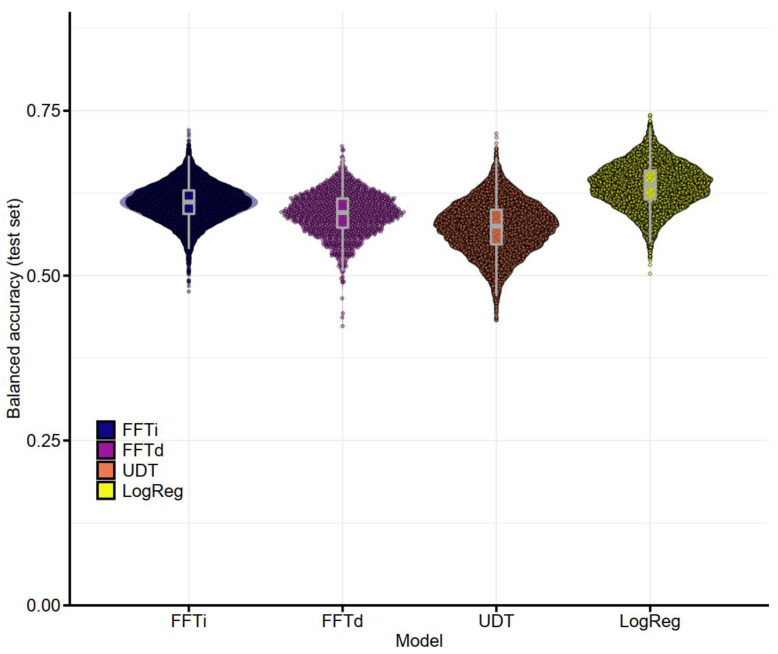
Balanced accuracy in the test set (prediction task) for the four algorithms across trials with pre-operative cues only. The graph shows boxplots and violin plots, with dots representing the results of individual trials (10,000 trials for FFTi, UDT and LogReg; 1000 trials for FFTd). FFTi—fast-and-frugal tree construction using the ifan algorithm, FFTd—fast-and-frugal tree construction using the dfan algorithm, UDT—unconstrained decision tree based on the CART algorithm, LogReg—logistic regression.

**Figure 3 jcm-11-05629-f003:**
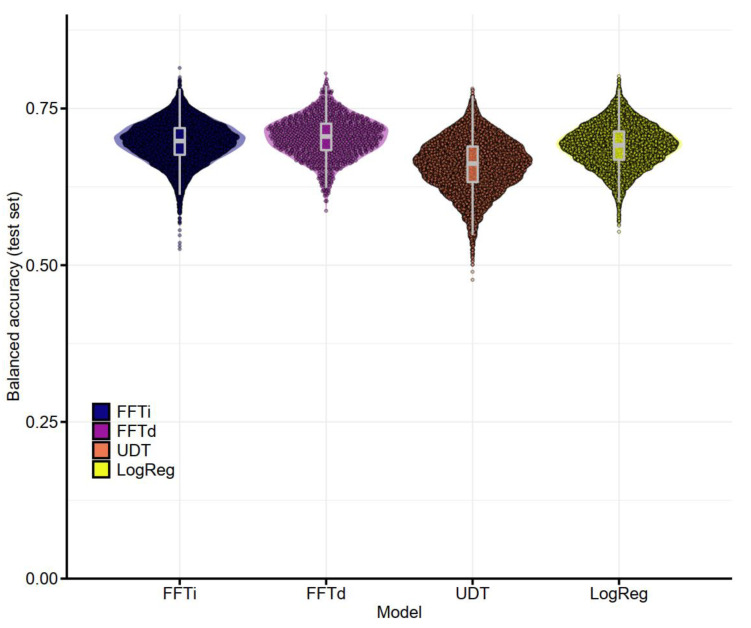
**Balanced accuracy in the test set (prediction task) for the four algorithms across trials with all cues**: The graph shows boxplots and violin plots, with dots representing the results of individual trials (10,000 trials for FFTi, UDT and LogReg; 1000 trials for FFTd). FFTi—fast-and-frugal tree construction using the ifan algorithm, FFTd—fast-and-frugal tree construction using the dfan algorithm, UDT—unconstrained decision tree based on the CART algorithm, LogReg—logistic regression.

**Figure 4 jcm-11-05629-f004:**
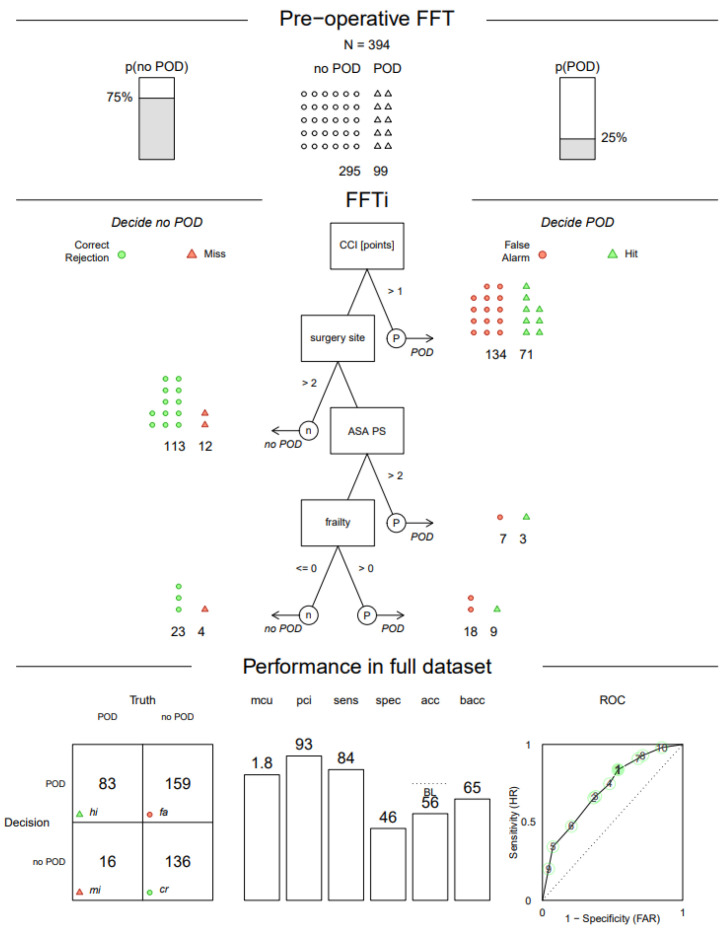
Pre-operative fast-and-frugal tree estimated with the ifan algorithm. POD—postoperative delirium, p(POD)—probability of POD a priori (base rate), p (no POD)—complement of p(POD), FFTi—fast-and-frugal tree construction using the ifan algorithm, CCI—Charlson Comorbidity Index, surgery site > 2—peripheral, ASA PS—physical status according to the American Society of Anesthesiologists, frailty <= 0—robust, frailty > 0—pre-frail/frail, mcu—mean cues used, pci—percent cues ignored, sens—sensitivity, spec—specificity, acc—unweighted accuracy, back—balanced accuracy (sensitivity + specificity)/2, BL—probable BL (the base rate of 75% that could be achieved by classifying all cases as negative), ROC—receiver operating characteristic (shows the performance of all compared trees using the same cue order numbered according to their resulting balanced accuracy (in the training set), each data point shows the false alarm rate (FAR) on the *x*-axis and sensitivity/hit rate (HR) on the *y*-axis), hi—hit, mi—miss, cr—correct rejection.

**Figure 5 jcm-11-05629-f005:**
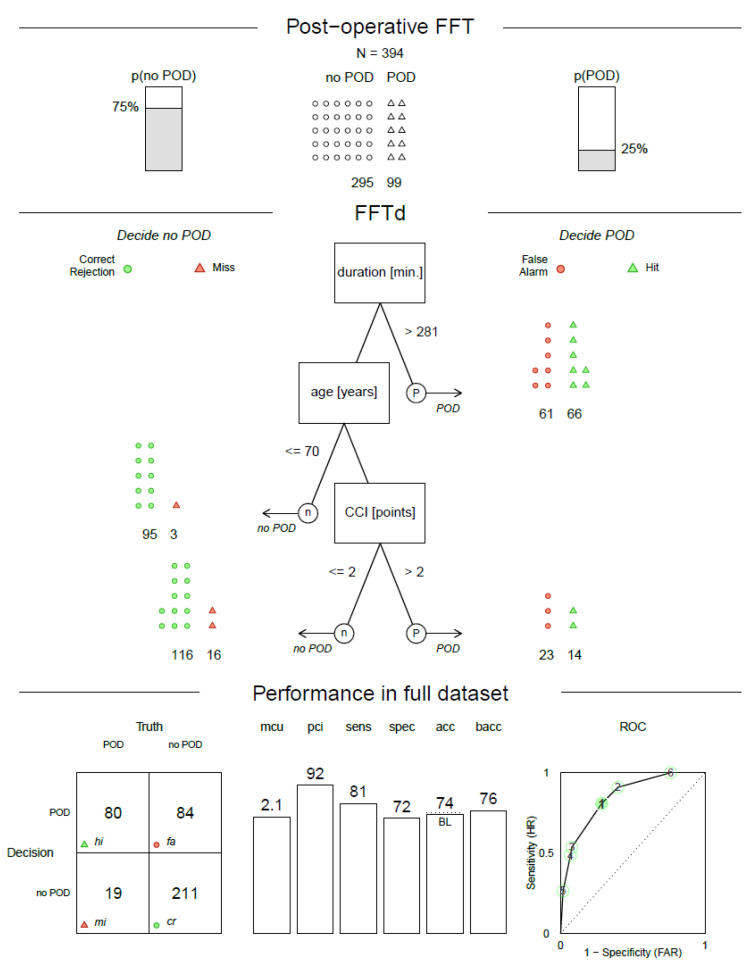
Postoperative fast-and-frugal tree estimated with the dfan algorithm. POD—postoperative delirium, p(POD)—probability of POD a priori (base rate), p (no POD)—complement of p(POD), FFTd—fast-and-frugal tree construction using the dfan algorithm, duration—duration of anesthesia, CCI—Charlson Comorbidity Index, mcu—mean cues used, pci—percent cues ignored, sens—sensitivity, spec—specificity, acc—unweighted accuracy, back—balanced accuracy (sensitivity + specificity)/2, BL—probable BL (the base rate of 75% that could be achieved by classifying all cases as negative), ROC—receiver operating characteristic (shows the performance of all compared trees using the same cue order numbered according to their resulting balanced accuracy (in the training set), each data point shows the false alarm rate (FAR) on the *x*-axis and sensitivity/hit rate (HR) on the *y*-axis), hi—hit, mi—miss, cr—correct rejection.

**Table 1 jcm-11-05629-t001:** Patient characteristics (n = 394).

Characteristic	POD (n = 99) 25.10%	Non POD (n = 295) 74.90%	*p*
	n = 394		
**Age (years)**	74 [71;77]	72 [68;76]	0.004 ^a^
**Sex**			
**Female**	51 (41.5%)	145 (49.2%)	0.684 ^b^
**ASA PS**			<0.001 ^b^
**1–2**	43 (43.4%)	203 (68.8%)	
**3–4**	56 (56.6%)	92 (31.2%)	
**Charlson Comorbidity Index**	2.14 ±1.5 ^c^	1.43 ± 1.6 ^c^	<0.001 ^a^
**Frailty status**			<0.001 ^b^
**Pre-frail**	50 (51.0%)	143 (49.1%)	
**frail**	30 (30.6%)	30 (10.3%)	
**Site of surgery**			<0.001 ^b^
**intracranial**	2 (2.0%)	7 (2.4%)	
**intrathoracic, intra-abdominal or pelvic**	67 (67.7%)	122 (41.4%)	
**peripheral**	30 (30.3%)	166 (56.3%)	
**Duration of Anaesthesia (min)**	360 [220;495]	157 [100;260]	<0.001 ^b^

Data are expressed as medians [25th quartile; 75th quartile] except for categorical data, which are expressed as frequencies (percentages). *p*-values are with respect to Mann–Whitney U test (^a^) or Chi-squared test (^b^) between patients with or without POD. Additionally, data for Charlson Comorbidity Index are presented as means ± SD (^c^). *p* ≤ 0.05 was considered as statistically significant. SD—standard deviation, POD—postoperative delirium, ASA PS—physical status according to the American Society of Anesthesiologists, min—minutes.

**Table 2 jcm-11-05629-t002:** **Average performance for algorithms across trials with pre-operative information.** The table presents means and standard errors separately for the four algorithms with the training set (fitting) and test set (prediction). For each combination, the table reports the mean and standard error of the mean for sensitivity, specificity and balanced accuracy across trials. FFTi, UDT and LogReg were tested in 10,000 trials and FFTd in the first 1000 of these only.

		Training			Prediction		
		Sensitivity	Specificity	Bal. Accuracy	Sensitivity	Specificity	Bal. Accuracy
FFTi	M	0.693	0.682	0.688	0.578	0.644	0.611
	SE	(0.0013)	(0.0013)	(0.0002)	(0.0015)	(0.0014)	(0.0003)
FFTd	M	0.751	0.689	0.720	0.562	0.625	0.593
	SE	(0.0037)	(0.0037)	(0.0007)	(0.0044)	(0.0040)	(0.0011)
UDT	M	0.868	0.738	0.803	0.52	0.626	0.573
	SE	(0.0006)	(0.0006)	(0.0002)	(0.0010)	(0.0007)	(0.0004)
LogReg	M	0.737	0.747	0.742	0.581	0.692	0.637
	SE	(0.0004)	(0.0003)	(0.0003)	(0.0007)	(0.0004)	(0.0003)

M—mean, SE—standard error of the mean, FFTi—fast-and-frugal tree construction using the ifan algorithm, FFTd—fast-and-frugal tree construction using the dfan algorithm, UDT—unconstrained decision trees based on the CART algorithm, LogReg—logistic regression.

**Table 3 jcm-11-05629-t003:** **Average performance of algorithms across trials with full information.** The table presents means and standard errors separately for the four algorithms with the training set (fitting) and test set (prediction). For each combination, the table reports the mean and standard error of the mean for sensitivity, specificity and balanced accuracy across trials. FFTi, UDT and LogReg were tested in 10,000 trials and FFTd in the first 1000 of these only.

		Training			Prediction		
		Sensitivity	Specificity	Bal. Accuracy	Sensitivity	Specificity	Bal. Accuracy
FFTi	M	0.767	0.723	0.745	0.698	0.695	0.696
	SE	(0.0008)	(0.0008)	(0.0002)	(0.0011)	(0.0009)	(0.0003)
FFTd	M	0.815	0.756	0.786	0.698	0.71	0.704
	SE	(0.0025)	(0.0024)	(0.0007)	(0.0032)	(0.0028)	(0.0010)
UDT	M	0.896	0.784	0.840	0.625	0.694	0.660
	SE	(0.0005)	(0.0005)	(0.0002)	(0.0010)	(0.0007)	(0.0004)
LogReg	M	0.792	0.803	0.798	0.632	0.749	0.690
	SE	(0.0004)	(0.0003)	(0.0003)	(0.0007)	(0.0004)	(0.0003)

M—mean, SE—standard error of the mean, FFTi—fast-and-frugal tree construction using the ifan algorithm, FFTd—fast-and-frugal tree construction using the dfan algorithm, UDT—unconstrained decision tree based on the CART algorithm, LogReg—logistic regression.

## Data Availability

Data available on request due to privacy restrictions.

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
