# Peer review of "Forecasting Postoperative Delirium in Older Adult Patients with Fast-and-Frugal Decision Trees"

_jcm, 2022, doi:10.3390/jcm11195629_

Round 1

Reviewer 1 Report

Dear Editor,

Dear Authors,

thank you for giving me an opportunity to review this manuscript. The aim of the study was to compare the capacity of Fast-and-Frugal Trees with compensatory models - Unconstrained Classification Trees and Logistic Regression in order to predict POD among older surgical patients in the perioperative setting.

The manuscript is well written, with adequate flow and is focused on an important clinical problem. I have some remarks that might be useful for the authors to improve the paper.

Line 40-42: Please report the incidence of POD with the support of relevant references, this may be shown as a range, but should be added. Not all readers are experts in the field. 

Line 46: The authors write that “screening for delirium should be performed once per shift in all patients”, however I would recommend enhancing this point and clarifying by adding “at least twice a day” and “for 5 days after surgery”. The shifts are different length, unfortunately, and in many places last for 24 hours. Monitoring once a day (once per shift for some places) is not enough.

Line 62: Is “particlarly”, should be “particularly”

Line 77-78: Please explain why both tools (Nu-DESC and CAM-ICU) and chart review were used? If we are aiming at making things simple this means extra work for any team.

Line 80: please verify the future tense used “we will build” and correct. The work has already been done.

Line 95: Please delete “see Error! Reference source not found”.

Line 97: Please provide the date of the Ethics Committee approval.

Line 117-118: Please explain (see Error! Reference source not found. for patient characteristics).

Table 1: The whole text from lines 124-130, other than the title, should be placed as a legend under the Table. Please explain all abbreviations in the Legend (POD, ASA).

Line 147: Is “Each”, should be “each”.

Line 292: Please explain all abbreviations in the Legend under Figure 2 (FFTi, FFTd, UDT).

Line 316: Please explain all abbreviations in the Legend under Figure 2 (FFTi, FFTd, UDT).

Line 412: Is “To the best of our knowledge, the first time decision trees were created for the risk stratification of POD. Should be: “To the best of our knowledge, this is the first time decision trees were created for the risk stratification of POD.

Line: Is: “perform similar well”, should be “perform similarly well”.

Line 491-502: The Conclusions are well written in the abstract and should be repeated here, with adequate re-phrasing or included as a whole “Based on the entire dataset, two different decision trees were developed for the pre-operative and post-operative settings. Within the pre-operative setting, FFTrees using four cues (Charlson Comorbidity Index (CCI), site of surgery, physical and frailty status) outperformed the more complex UDT algorithm with respect to their balanced accuracy, nearing the prediction level of the Logistic Regression. Within the post-operative setting, FFTrees containing only three cues (duration of anesthesia, age and CCI) outperformed both complex models. Given that both FFTrees contain considerably fewer criteria, which can be easily memorized and applied by health professionals in daily routine, FFTrees could help identify patients requiring intensified POD screening.”

With best regards

Author Response

Reviewer 1

Dear Authors,

thank you for giving me an opportunity to review this manuscript. The aim of the study was to compare the capacity of Fast-and-Frugal Trees with compensatory models - Unconstrained Classification Trees and Logistic Regression in order to predict POD among older surgical patients in the perioperative setting.

The manuscript is well written, with adequate flow and is focused on an important clinical problem. I have some remarks that might be useful for the authors to improve the paper.

Thank you for taking the time to review our manuscript and for your valuable comments. The manuscript has been revised carefully and we believe that the quality has been considerably increased.

Line 40-42: Please report the incidence of POD with the support of relevant references, this may be shown as a range, but should be added. Not all readers are experts in the field.

Thank you for this important advice. We added information on incidence of POD.

Line 46: The authors write that “screening for delirium should be performed once per shift in all patients”, however I would recommend enhancing this point and clarifying by adding “at least twice a day” and “for 5 days after surgery”. The shifts are different length, unfortunately, and in many places last for 24 hours. Monitoring once a day (once per shift for some places) is not enough.

We fully agree with you and have adjusted this paragraph.

Line 62: Is “particlarly”, should be “particularly”

Thank you for pointing this out. We adjusted the typo.

Line 77-78: Please explain why both tools (Nu-DESC and CAM-ICU) and chart review were used? If we are aiming at making things simple this means extra work for any team.

Thank you for your comment. We have removed this information in the introduction in favor of clarity. Assessment of POD using Nu-DESC and CAM-ICU in addition to patient chart review was performed in the BioCog study to secure high quality assessment of the primary endpoint “POD” (see section 2.2. of the manuscript). Building of FFTress was based on the dataset of the BioCog study. Application of the FFTrees does not contain assessment of POD using tools such as Nu-DESC or CAM-ICU or a chart review.

Line 80: please verify the future tense used “we will build” and correct. The work has already been done.

Thank you for pointing this out. We have adjusted this accordingly.

Line 95: Please delete “see Error! Reference source not found”.

Thank you for pointing this out. We have adjusted this accordingly.

Line 97: Please provide the date of the Ethics Committee approval.

Thank you for pointing this out. We have added the date of Ethics Committee approval.

Line 117-118: Please explain (see Error! Reference source not found. for patient characteristics).

Thank you for pointing this out. We have adjusted this accordingly.

Table 1: The whole text from lines 124-130, other than the title, should be placed as a legend under the Table. Please explain all abbreviations in the Legend (POD, ASA).

Thank you for pointing this out. We have adjusted this accordingly

Line 147: Is “Each”, should be “each”.

Thank you for pointing this out. We adjusted the typo.

Line 292: Please explain all abbreviations in the Legend under Figure 2 (FFTi, FFTd, UDT).

Thank you for pointing this out. We have adjusted this accordingly.

Line 316: Please explain all abbreviations in the Legend under Figure 2 (FFTi, FFTd, UDT).

Thank you for pointing this out. We have adjusted this accordingly.

Line 412: Is “To the best of our knowledge, the first time decision trees were created for the risk stratification of POD. Should be: “To the best of our knowledge, this is the first time decision trees were created for the risk stratification of POD.

Thank you for this good advice. We have adjusted this accordingly.

Line: Is: “perform similar well”, should be “perform similarly well”.

Thank you for pointing this out. We adjusted the typo.

Line 491-502: The Conclusions are well written in the abstract and should be repeated here, with adequate re-phrasing or included as a whole “Based on the entire dataset, two different decision trees were developed for the pre-operative and post-operative settings. Within the pre-operative setting, FFTrees using four cues (Charlson Comorbidity Index (CCI), site of surgery, physical and frailty status) outperformed the more complex UDT algorithm with respect to their balanced accuracy, nearing the prediction level of the Logistic Regression. Within the post-operative setting, FFTrees containing only three cues (duration of anesthesia, age and CCI) outperformed both complex models. Given that both FFTrees contain considerably fewer criteria, which can be easily memorized and applied by health professionals in daily routine, FFTrees could help identify patients requiring intensified POD screening.”

Thank you for this very good suggestion. We have adjusted this accordingly.

Reviewer 2 Report

The authors support the use of Fast-and-Frugal decision trees to identify postoperative delirium.  Comparisons were made to several other algorithms. 

The issue that I have is that the manuscript is extremely difficult to follow beginning with the abstract.  The Materials and Methods section is quite long with six sub-sections.  Throughout this section, as well as in the Results, the text rambles with sentences that are more appropriate for the Discussion. 

This manuscript needs to be totally revised.  It is difficult to follow because the Materials and Methods is too long to understand what was done.  Both the Methods and Results have extensive verbiage that should be in the Discussion.  The abstract should be a clear snapshot of the study and easy to read at a glance. 

Author Response

Reviewer 2

The authors support the use of Fast-and-Frugal decision trees to identify postoperative delirium.  Comparisons were made to several other algorithms.

The issue that I have is that the manuscript is extremely difficult to follow beginning with the abstract.  The Materials and Methods section is quite long with six sub-sections.  Throughout this section, as well as in the Results, the text rambles with sentences that are more appropriate for the Discussion.

This manuscript needs to be totally revised.  It is difficult to follow because the Materials and Methods is too long to understand what was done.  Both the Methods and Results have extensive verbiage that should be in the Discussion.  The abstract should be a clear snapshot of the study and easy to read at a glance.

Thank you for taking the time to review our manuscript. Your advice on comprehensibility was of high importance. We have taken up your points and adapted the abstract, shortened the methodology section and put some passages in the supplement. According to your suggestions, we have critically reviewed the methods and results section and transferred appropriate points to the discussion (please see tracked changes). In addition, we had non-involved colleagues read the manuscript to identify further points of difficulty in comprehensibility.

We very much hope that we have been able to improve the manuscript and that it is now possible to follow the content without any problems. If you notice any other aspects, we would be very pleased if you could make concrete suggestions.

Round 2

Reviewer 2 Report

The authors have greatly improved the manuscript.  It reads much better compared to the original.